# 3D printing of bioreactors in tissue engineering: A generalised approach

Marius Gensler[1]*, Anna Leikeim[1], Marc Möllmann[2], Miriam Komma[1], Susanne Heid[3], Claudia Müller[4], Aldo R. Boccaccini[3], Sahar Salehi[4], Florian Groeber-Becker[1,2], Jan Hansmann[1,5]

1 Department Tissue Engineering and Regenerative Medicine, University Hospital Würzburg, Würzburg, Germany, 2 Translational Center Regenerative Therapies, Fraunhofer Institute for Silicate Research, Würzburg, Germany, 3 Institute of Biomaterials, University of Erlangen-Nürnberg, Erlangen, Germany, 4 Department Biomaterials, University of Bayreuth, Bayreuth, Germany, 5 Faculty of Electrical Engineering, University of Applied Sciences Würzburg-Schweinfurt, Schweinfurt, Germany

☯ These authors contributed equally to this work.
* marius.gensler@uni-wuerzburg.de

**Data Availability Statement:** All relevant data are within the paper and its Supporting information files.

**Funding:** This study is funded by the Deutsche Forschungsgemeinschaft (DFG, German Research

## Abstract

3D printing is a rapidly evolving field for biological (bioprinting) and non-biological applications. Due to a high degree of freedom for geometrical parameters in 3D printing, prototype printing of bioreactors is a promising approach in the field of Tissue Engineering. The variety of printers, materials, printing parameters and device settings is difficult to overview both for beginners as well as for most professionals. In order to address this problem, we designed a guidance including test bodies to elucidate the real printing performance for a given printer system. Therefore, performance parameters such as accuracy or mechanical stability of the test bodies are systematically analysed. Moreover, post processing steps such as sterilisation or cleaning are considered in the test procedure. The guidance presented here is also applicable to optimise the printer settings for a given printer device. As proof of concept, we compared fused filament fabrication, stereolithography and selective laser sintering as the three most used printing methods. We determined fused filament fabrication printing as the most economical solution, while stereolithography is most accurate and features the highest surface quality. Finally, we tested the applicability of our guidance by identifying a printer solution to manufacture a complex bioreactor for a perfused tissue construct. Due to its design, the manufacture via subtractive mechanical methods would be 21-fold more expensive than additive manufacturing and therefore, would result in three times the number of parts to be assembled subsequently. Using this bioreactor we showed a successful 14-day-culture of a biofabricated collagen-based tissue construct containing human dermal fibroblasts as the stromal part and a perfusable central channel with human microvascular endothelial cells. Our study indicates how the full potential of biofabrication can be exploited, as most printed tissues exhibit individual shapes and require storage under physiological conditions, after the bioprinting process.

## Introduction

In 1993, Langer and Vacanti introduced the term "tissue engineering" (TE) as an interdisciplinary field that applies the principles of engineering and the life sciences toward the

Foundation) – Projectnumber 326998133 – TRR 225 (subproject B03).

**Competing interests:** The authors indicate no conflict of interest.

development of biological substitutes that restore, maintain, or improve tissue function [1]. Since then, various successful attempts showing the feasibility of the tissue-engineered biological substitutes, were achieved for bladder tissue [2–4], artificial arteries made by patient's stem cells [5–7], trachea [8–10], meniscus [11] and skin [12,13], just to name a few. In TE, cells, material and growth-stimulating signals are combined to generate viable and functional tissues [14–17]. By their self-renewal and multipotent properties, stem cells, which are used in many approaches, allow to build up complex constructs [18–21]. Alternatively, patient-derived somatic cells enable generation of personalised grafts and the reduction of the risk of rejection of the engineered transplant [18,22]. As cells grow in a 3-dimensional (3D) microenvironment *in vivo*, suitable scaffolds are required for the proper organisation of generated tissues *in vitro* [15,16]. These scaffolds exhibit distinct properties like biocompatibility, cell adhesion and pore size and can be of artificial or biological origin [15,17,22]. Collagen is often used as it represents the most common protein in human connective tissue providing strength and structural stability [15]. In addition to a 3D microenvironment, culture conditions are a crucial factor for a proper tissue development and growth.

To provide and maintain tissue-specific process conditions, engineered tissues are often cultured in s- called bioreactor systems [23]. Common for all bioreactors is the capability to store a tissue construct while simulating the natural environmental conditions in the most physiological way [23]. For example, bioreactors mimic blood flow by perfusing the tissue construct with the appropriate cell culture medium. This approach also improves the supply of nutrients. Additionally, perfusion-induced shear stress supports maintenance of the physiological state for many cell types such as endothelial cells. Beyond these flow-induced mechanical stimuli, complex bioreactors are also able to apply additional signals such as mechanical loading or torsion to the tissue to increase the correlation to natural conditions [24]. As a further general requirement, a bioreactor protects the construct from contaminants. Importantly, the whole bioreactor has to be made from biocompatible materials that do not leach harmful substances. Further requirements and general concepts of bioreactor design have been discussed in the literature [23,25]. Until now, the gold standard for manufacturing bioreactor components are subtractive manufacturing methods like milling, drilling, grinding etc. as well as injection molding or welding and punching [25]. However, milling methods are very costly, time intensive and high throughput is not efficient. On the other hand, high throughput molding and punching requires a durable mold, which is not adaptable following production and thereby not suitable for prototyping. This however, is possible if the components are manufactured by grinding processes. Further disadvantages of standard subtractive manufacturing methods are limitations in the technical degree of freedom, leading to non-producible geometries, as well as long manufacturing times.

To overcome these limitations, attention is focused on modern additive manufacturing processes, such as 3D printing. These techniques are well suited for rapid prototyping of complex organic shapes, as they can be found in nature, and hollow geometries [26]. In general, 3D printing excels conventional processing methods in average production time and cost, as well as post processing, waste production and resource consumption [27]. Moreover, it significantly decreases the amount of single pieces required for the same structure resulting in less leakage. First a 3D model is designed by computer aided design (CAD) software and specifically sliced into layers by a slicer software. Within the slicing step a G-code or G-code-like file is generated for opperating the printer device to print the desired geometry. The three most applied printing methods are fused filament fabrication (FFF), stereolithography (SLA) and selective laser sintering (SLS). Moreover, there are more techniques available, which are officially summarised and categorised in DIN EN ISO/ASTM 52900:2018 [28]. In FFF, a

continuous filament of a defined diameter is fed into the printhead, where it is melted. The high viscous material is then pressed through a nozzle onto the print bed and immediately resolidified by cooling down. The printhead is moved in X and Y direction relative to the print bed, which is moved in Z direction. Layer after layer the designated geometry is built up. SLA instead uses liquid materials for printing. Here, a photo crosslinkable polymer or resin is filled into a translucent chamber. In contrast to FFF, the print bed in SLA is upside down and drives in Z direction through the liquid material onto the bottom of the chamber. A light-source crosslinks the resin through the chamber bottom. After each layer is crosslinked, the print bed with the part under manufacturing attached to it moves up and the material in the chamber gets equally distributed before the next layer is added. For SLS, fine powders are used for printing. SLS printers usually have two chambers. First, a printing chamber gets filled with a thin layer of the powder material before a laser melts the particles together. When one layer is completed, a new layer of powder is set on top of the previous and the process is repeated. 3D printing is already well established in fields like art [29,30], fashion [31,32], architecture [29,33] and car industry [34,35]. Even crucial parts in the aerospace industry are produced by 3D printing [27,36–38].

Interestingly, 3D printing has also found its way into medical technology, recently [39–41]: prostheses and implants are manufactured this way but also models for surgical planning and training [42–47]. The advantages of 3D printing can be fully exploited here, as specially tailored solutions for an individual patient can easily be implemented. The high flexibility of additive manufacturing is also advantageous for applications in TE. For example, 3D printing techniques could enable the production of uniquely designed bioreactors with individual tissue and patient specifications. To create a printed bioreactor, the printed material has to meet the needs for bioreactors as mentioned above, first of all biocompatibility, serializability and no leakage. Although the application of 3D printing increases in industry, research and even private use, there are only few studies published that harnessed 3D printed bioreactors. An example for a 3D printed bioreactor system was presented in 2018 by Smith et al. with the FABRICA bioreactor [48].

Nevertheless, literature research in PubMed for the terms "3D printing", "bioreactor" and "tissue engineering" has only resulted in about 7 publications concerning 3D printed bioreactors or bioreactor-like chambers (data request June 22th, 2020). We see the reason for this is in the high variety of materials, printers and techniques, which are hard to grasp by research groups that do not have access to extensive knowledge in material and engineering science.

The article presented here aims to provide guidance for using additive manufacturing methods in regenerative medicine and TE. Therefore, the three most prominent printing techniques FFF, SLA, and SLS were compared and specific capabilities and limitations were derived. A focus for the assessment of the printing method was made on limits for small and detailed geometries, as these kinds of structures usually cause exponentially increasing manufacturing effort, time and costs when using standard subtractive methods. Additionally, we tested the impact of post processing like sterilisation on the used material. From the study results, a guidance for the measurement of the real printer performance is derived. The developed guidance for the assessment of the printers was applied in an interlaboratory evaluation to ensure the robustness and usability of the test procedure. By using our presented standardised procedure, an individual 3D printer can be characterised according to its capacity of printing bioreactor systems for TE applications. Therefore, the test geometries are made available online and after printing those, required parameters for printing a bioreactor can be systematically assessed using the methods described in this article. Finally, the guidance was applied to identify the most suitable method for the manufacturing of a complex bioreactor system for culturing a vascularized 3D human tissue to prove the applicability of the test procedure described here.

## Materials and methods

### CAD design

Computer-aided design (CAD) of the geometries to be printed was performed with Solidworks® Premium 2017 (Dassault Systèmes, France) and converted into STL-files.

### 3D printing and post processing

Machine-specific G-code-like files were generated from the STL-files for every printer using the individual company own software for the respective printer. SLA-printed parts were washed in isopropanol (Carl Roth GmbH, Germany) for 5 min using the Form Wash device (Formlabs Inc., USA). Afterwards, the parts were cured with UV-light at 60°C for 30 min using the Form Cure device (Formlabs Inc., USA). Subsequently, SLS-printed parts were manually sand blasted for a few seconds using the Sinterit Sandblaster (Sinterit sp. z o.o., Poland). FFF-printed parts were not post processed after printing. All important information to printers and printing parameters are listed in Table 1. Raise3D Pro 2 printer was used as FFF control.

### Analysis of accuracy and leakage

For analysing the X, Y and Z resolution, as well as the feret diameter (largest diameter) and the roundness of the channels, pictures were taken with a light microscope (EVOS XL, Thermo Fisher Scientific Inc., USA) and then parameters were quantified using self-written macros within Fiji (ImageJ) [49]. Angles were measured using the angle measuring tool implemented in Fiji. For leakage quantification, printed cups were filled with 1 ml deionised water and

**Table 1. Devices, materials and printing parameters for 3D printing.**

| Printer | Raise3D Pro 2 | Ultimaker 3 | Ultimaker S5 | Form 2 | Lisa Pro |
|---|---|---|---|---|---|
| Method | FFF | FFF | FFF | SLA | SLS |
| Software (file format) | ideamaker 3.4.2 (.gcode) | Ultimaker Cura 4.2.1 (.gcode) | Ultimaker Cura 4.3 (.gcode) | Preform 3.0.1 (.form) | Sinterit Studio 2019 1.4.5.0 (.scode) |
| Company (printer) | RAISE3D Technologies, USA | Ultimaker B.V., Netherlands | Ultimaker B.V., Netherlands | Formlabs Inc., USA | Sinterit sp. z o.o., Poland |
| Printed material | Green-TEC Pro Filament, Natur, 1.75 mm | Green-TEC Pro Filament, Natur, 2.85 mm | Green-TEC Pro Filament, Natur, 2.85 mm | Dental SG Resin | PA12 Smooth |
| | (Lignin based*) | (Lignin based*) | (Lignin based*) | (Bisphenol A ethoxylate based*) | (Nylon 12) |
| Company (material) | Extrudr \| FD3D GmbH, Austria | Extrudr \| FD3D GmbH, Austria | Extrudr \| FD3D GmbH, Austria | Formlabs Inc., USA | Sinterit sp. z o.o., Poland |
| Basic printing parameter | Nozzle diameter: 0.4 mm | Nozzle diameter: 0.4 mm | Nozzle diameter: 0.4 mm | Laser diameter: 140 μm | |
| | Layer height: 200 μm | Layer height: 100 μm | Layer height: 100 μm | Layer height: 50 μm | Layer height: 125 μm |
| | Printing temp: 220°C | Printing temp: 220°C | Printing temp: 220°C | Printing temp: 30.5°C | Chamber temp: 178°C |
| | Printbed temp: 60°C | Printbed temp: 60°C | Printbed temp: 60°C | Laser power: 0.25 W | Laser power: 1.67 W |
| | Standard speed: 50 mm/s | Standard speed: 70 mm/s | Standard speed: 70 mm/s | | |
| | Outer shells: 4 | Outer shells: 8 | Outer shells: 4 | | |
| | Infill ratio: 33% | Infill ratio: 20% | Infill ratio: 15% | | |
| | Infill type: Honeycomb | Infill type: Honeycomb | Infill type: Triangle | | |

Fused filament fabrication (FFF), stereolithography (SLA), selective laser sintering (SLS), temperature (temp).

*Since materials for printing are complex composites, we only show the main component here.

incubated for 24 h in a closed petri dish. The weight of the cups was measured before and after filling as well as after incubation. The remaining water was then calculated.

## Sterilisation methods

The autoclaving process was performed at 121˚C for 15 min using the autoclaving device DX-45 (Systec GmbH, Germany). Including heating and cooling steps, the whole process took up to 1.5 h.

For vaporised hydrogen peroxide ($VH_2O_2$) plasma sterilisation, the chamber of a plasma cleaner device (Pico LF PC 115656, Diener electronic GmbH & Co. KG, Germany) was pre-heated by an induced oxygen plasma with a generator power of 500 W for 12 min at a pressure of 0.3 mbar created by a gas flow of 12 standard cubic centimeters per minute (sccm). Directly after the heating process, the foil-wrapped (Stericlin, VP Group, Germany) 3D printed parts and a metal vaporiser unit filled with 1.5 ml of 60% $H_2O_2$ solution (Thermo Fisher Scientific, USA) were placed in the plasma chamber. After evacuation to 4.0 mbar, the printed parts were kept in that atmosphere for 75 min before the chamber was evacuated to 0.4 mbar. In this low-pressure $H_2O_2$ atmosphere, the parts were plasma treated at 300 W for 4 min. In a subsequent step, the atmosphere in the chamber was replaced by pure $O_2$ gas to a pressure of 0.4 mbar before the start of the last plasma process at 300 W for 5 min. The chamber was then flooded by pure oxygen gas for 120 s, before the atmosphere in the chamber was finally replaced by air.

## Mechanical analysis

For determining the mechanical properties, tension and bending tests were performed using a material testing machine (Z010, ZwickRoell GmbH & Co. KG, Germany). All tests were conducted at a speed of 5 mm/min with a 10 kN measuring unit. For the four-point bending test, the support span was set to 40 mm and the span length to 10 mm. Tensile and flexural strength were calculated from the measured data using the related machine software testXpert II V3.1 (ZwickRoell GmbH & Co. KG, Germany). Both testing methods were performed referring to EN ISO 527 [50] and 178 [51].

## Cell isolation and culture

Primary cells were isolated from human juvenile foreskin biopsies with approval of the local ethical board of the University of Würzburg (vote 182/10) and with the confirmed consent of their legal representatives. Isolation was performed according to previously published protocols [52,53]. Following, human dermal fibroblasts (hdF) were cultured in Dulbecco's modified Eagle medium (DMEM, Life Technologies, USA) with 10% fetal bovine serum (Biochrom AG, Germany) supplemented with 1% penicillin/streptomycin (Life Technologies, USA). Human dermal microvascular endothelial cells (hdmEC) were cultured in VascuLife® VEGF-Mv (Lifeline Cell Technologies, USA) supplemented with the dedicated LifeFactors Kit. Instead of the optional antimicrobial supplement, the medium was supplemented with 1% penicillin/streptomycin. For both cell types, medium was exchanged every two to three days. The cells were kept in a 95% humidified incubator at 37˚C and 5% $CO_2$. HdFs and hdmECs were used for experiments from passage 2 to 5 at a confluency of 80 to 90%.

## Casting of collagen hydrogel and dynamic culture

Collagen hydrogels were generated based on a previously published protocol [54]. Briefly, collagen type I from rat-tail at 8 to 10 mg/ml in 0.1% acetic acid was mixed with gel neutralisation solution (GNS) at a 2:1 volumetric ratio. HdFs were added to the GNS to a final

concentration of $4 \times 10^5$ cells/ml hydrogel before mixing. The mixed hydrogel was then filled into the assembled bioreactor. To generate a primary channel-like structure for perfusion, a cannula was inserted in the bioreactor beforehand. After polymerisation, the hydrogel was cultured for 24 h at 37˚C. Next, the cannula was removed, thereby leaving a central channel for perfusion. Subsequently, the bioreactor was connected to a fluidic circuit containing a medium reservoir filled with 30 ml of VascuLife® medium. Medium flow was set to a constant speed of 2.67 ml/min. The next day, the channel was seeded with $2 \times 10^6$ hdmECs in 250 µl VascuLife® medium. After 4 h of static incubation, the bioreactor was reconnected to the fluidic circuit with fresh VascuLife® medium. The flow velocity was decreased to 0.89 ml/min and again increased daily by 0.89 ml/min up to a maximum volume flow of 2.67 ml/min. The hydrogel was cultured in the bioreactor and perfused under constant flow conditions for up to 14 days. The medium was exchanged weekly.

## Viability assessment

To verify the viability of the cells in the hydrogel, metabolic activity was examined by performing a 3-(4,5-dimethylthiazol-2-yl)-2,5-diphenyltetrazoliumbromid (MTT) assay. After 14 days of culture, tissue constructs were removed from the bioreactor and placed into 2 ml of MTT solution (1 mg/ml; Sigma-Aldrich, USA). After incubationof 4 h at 37˚C, remaining MTT solution was discarded and the stained hydrogels were assessed optically.

## Biocompatibility test

Modified from ISO 10993–5 [55], quadratic platelets with an edge length of 17.3 mm and a thickness of 1 mm were printed with the investigated material. The platelets were incubated overnight in 1 ml DMEM (Life Technologies, USA) supplemented with 1% penicillin/streptomycin (Life Technologies, USA). 20000 hdFs per well were seeded in a 96-well plate in normal culture medium. 24 h after seeding, the medium was changed to the platelet-conditioned medium supplemented with 10% fetal bovine serum (Biochrom AG, Germany). Normal cell culture medium served as positive control and medium containing 1% sodium dodecyl sulfate (SDS, Bio-Rad Laboratories GmbH, Germany) as negative control. After another 24 h, a viability assay was performed: The medium was aspirated and 200 µl PBS+ (Sigma-Aldrich, USA) containing 1% water soluble tetrazolium (WST-1, Roche Holding GmbH, Germany) was applied for 60 min. Absorption was measured at 450 nm using a spectrophotometer (Infinite 200M, Tecan Trading AG, Switzerland). The material was stated as biocompatible if values were equal or higher than 70% referring to the positive control.

## Histological analysis

Hydrogels were removed from the bioreactors, fixed overnight at 4˚C using Histofix® (Carl Roth GmbH, Germany) and embedded in paraffin (Carl Roth GmbH, Germany). Cross sections of 3.5 µm thickness were stained with hematoxylin & eosin (HE; Morphisto GmbH, Germany) for a general anatomic overview.

Immunochemical stainings were performed using the SuperVision 2 HRP Kit (DCS Innovative Diagnostik-Systeme, Germany) following the manufacturer's instructions. Briefly, deparaffinized sections were treated with 3% $H_2O_2$ to block endogenous peroxidases and subsequently incubated with the primary antibody against Platelet endothelial cell adhesion molecule (mouse anti-CD31, 1:100, Agilent Dako, USA) for 1 h. After labelling with a horseradish-peroxidase polymer, 3,3'-diaminobenzidine was added as substrate, which is oxidised to a brownish dye. To visualise the tissue structure, counterstaining with hematoxylin was performed.

For immunofluorescence staining, sections were deparaffinized and blocked using 5% BSA in PBS- (Sigma-Aldrich, USA) containing 0.03% Triton-X-100 (Sigma-Aldrich, USA) for 20 min at RT. Samples were incubated with primary antibodies (mouse anti-CD31, 1:100, Agilent Dako, USA; rabbit anti-vimentin, 1:1000, Abcam, United Kingdom) overnight at 4˚C. Slides were washed and subsequently incubated with fluorophore-conjugated secondary antibody (donkey anti-mouse Alexa Fluor 555 and donkey anti-rabbit Alexa Fluor 488, 1:400, Thermo Fisher Scientific, USA) for 1 h at RT. After repeated washing, slides were covered with Fluoromount-DAPI (Thermo Fisher Scientific, USA). Microscopic images were acquired using a Keyence microscope Biorevo BZ-9000 (Keyence Corporation, Japan).

## Statistical analysis

Presented data is shown as mean ± standard deviation. Calculation was done in GraphPad Prism 6 (GraphPad Software, USA) and Excel 2016 (Microsoft Corporation, USA). Significances between groups were calculated by unpaired t-test. Significances are indicated by * if $p \leq 0.05$.

## Results

### Standardised quantification of CAD designed test bodies

To compare 3D printing methods and specific printers of the respective method, test bodies were designed to assess X, Y and Z resolution, printing of channels, angled overhangs and squared cups, which were used for measuring the leakage of the printed parts. Each of the test bodies was designed with multiple dimensions of the analysed parameter as shown in Fig 1. The length of the printed channels was 10 mm. The geometry of the test bodies was optimised for visualy assessing the shape with a microscope. Unlike other test bodies combining all parameters in a complex part, we were using individual test bodies for each parameter for independent assessment.

The test bodies were designed in Solidworks software, therefore containing the most digital information possible in the company's own SLDPT-format (Fig 2A). By transferring these files into STL-format, the surfaces were converted into a set of triangles (Fig 2B), which showed no impact to the geometry volume in Solidworks. By transferring the file into G-code, the path of the nozzle during printing could be visualised, showing that the resolution of printing in FFF technique is about 200 µm, thus smaller features couldn't be printed (Fig 2C). By generating the G-code-like format within the SLA and SLS printers software, a scheme of the laser path indicated that both printing methods had higher resolution than FFF and did not ignore any part of the geometry (Fig 2D and 2E). However, SLS software demonstrated that there was a certain precision offset from laser path to geometry (Fig 2E).

Further loss of accuracy occured due to the printing process itself, the physical behavior of the printed material and the post processing procedure (Fig 2F–2H). To quantify the resulting accuracy of the printed parts, two distinct Fiji macros were written (S1 Fig) for analysing the X, Y and Z resolution test bodies (macro XYZ) and for analysing the channel diameter and roundness (macro channels). Firstly, the area of interest was cropped and set to 8-bit grayscale. Subsequently, an unsharp mask filter was applied to increase contrast, followed by a grey level threshold (RenyiEntropy) to reduce noise. At the end, the particle analyser tool identified the shape of the area of interest (Fig 3). SLA printed parts in fact showed increased artifacts due to irregular light conditions caused by transparency of the material. Therefore, the RenyiEntropy filter had to be adjusted individually.

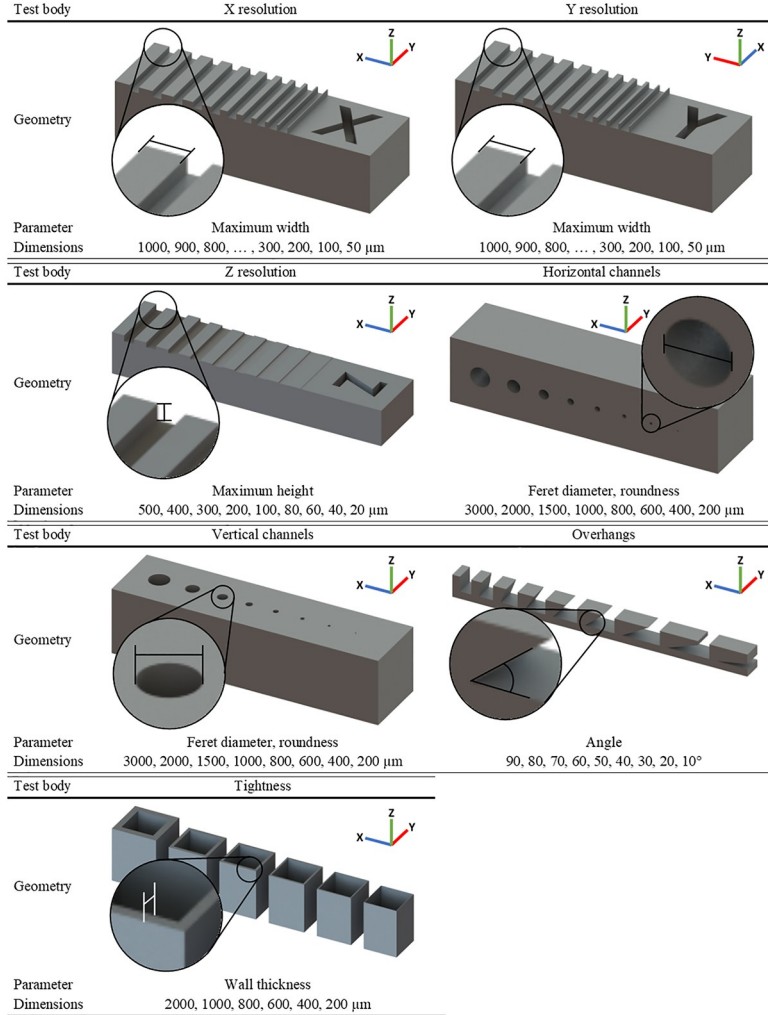

**Fig 1. Test bodies and parameters for 3D printing comparison.** Individual test bodies shown for every analysed parameter and orientation within the printers.

## Printing limits showing either oversized printing or no printing

Test bodies were printed by FFF, SLA and SLS and analysed for their accuracy (Fig 4). In general, positive deviation values indicate geometries that turned out thicker and negative values indicate geometries that turned out smaller than designed.

Comparing X and Y resolution (Fig 4A), all three techniques showed exponential increase of accuracy deviation by decreasing parameter dimensions of the test bodies. Thereby, SLA achieved the lowest deviation compared to FFF and SLS, especially in dimensions smaller than 500 μm. While FFF did not print dimensions below 200 μm at all, SLA and SLS printed every structure of the test body, resulting in very high deviations for small dimensions. Especially in SLS, the resulting geometries were printed so thick that they fused into each other (Fig 2H), which made it impossible to analyse them below 500 μm. By analysing the Z resolution, SLA also had the lowest deviation compared to FFF and SLS. Inaccuracy increased exponentially in heights below 100 μm, resulting in an oscillating deviation of 28.3 ± 4.8% at

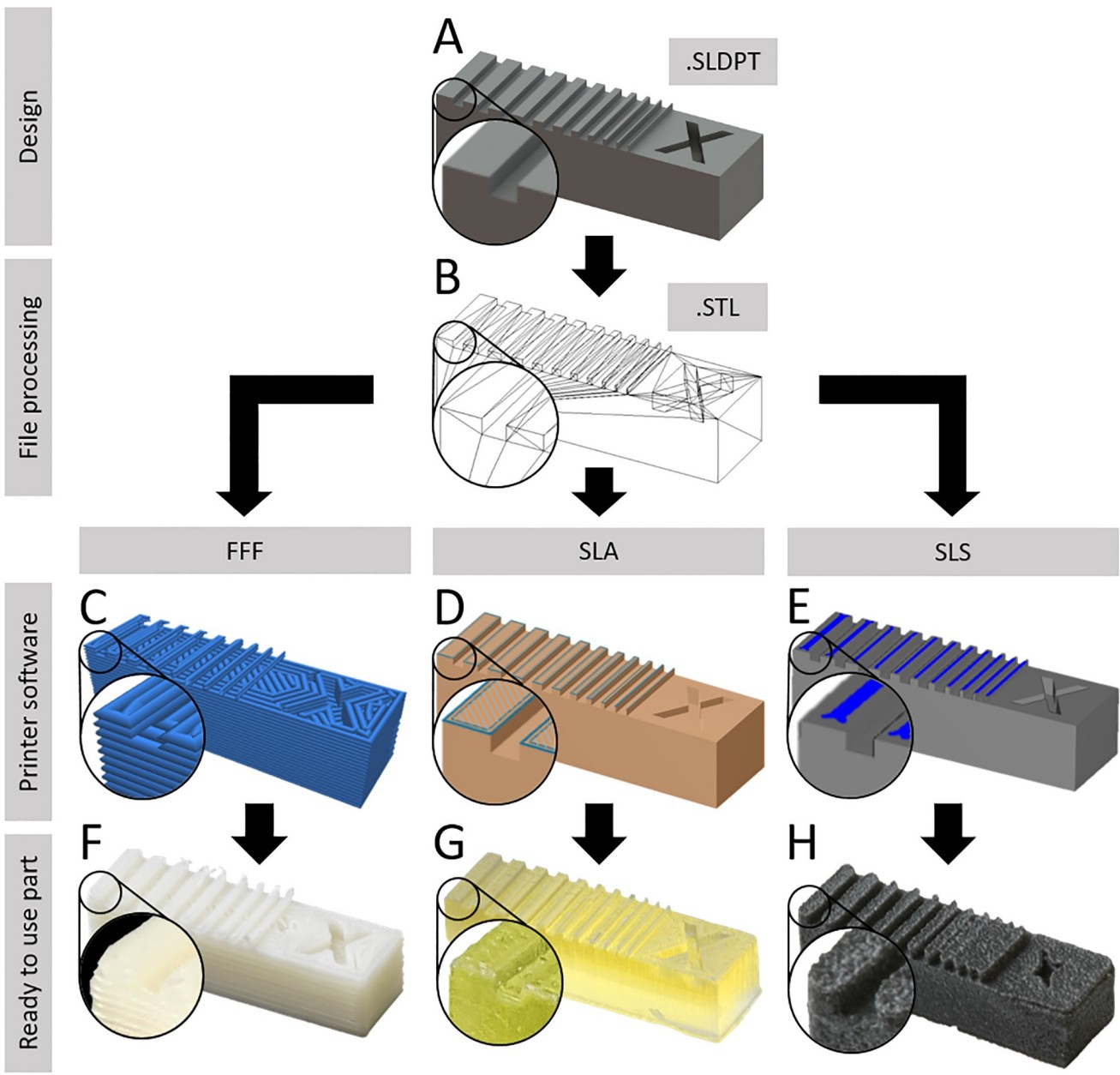

**Fig 2. Processing of CAD files to printed parts (example X-resolution test body).** All test bodies were designed in Solidworks 2017 software (A) and converted into STL-file format (B). The STL-file was then processed into the device-specific G-codes using the respective company owned software for the FFF printers (C), SLA printer (D) and the SLS printer (E). The resulting printed part after post processing (ready to use) shows significant differences in surface quality and accuracy (F-H). The nozzle or laser path is shown in blue.

80 μm, 6.1 ± 12.7% at 60 μm and 57.3 ± 31.8% at 40 μm height. Heights below 40 μm were not printed by SLA. FFF did not print below 100 μm and SLS stopped printing at 80 μm.

As SLS showed the lowest accuracy in the base axes, horizontal channels (Fig 4B) were only printable down to 2000 μm diameter. SLA achieved channels down to 800 μm diameter and FFF down to 400 μm. SLA-printed channels had a stable average deviation of 5.2 ± 0.8% from 3000 to 1000 μm diameter and then falling to -8.0 ± 7.6% at 800 μm. FFF instead showed a

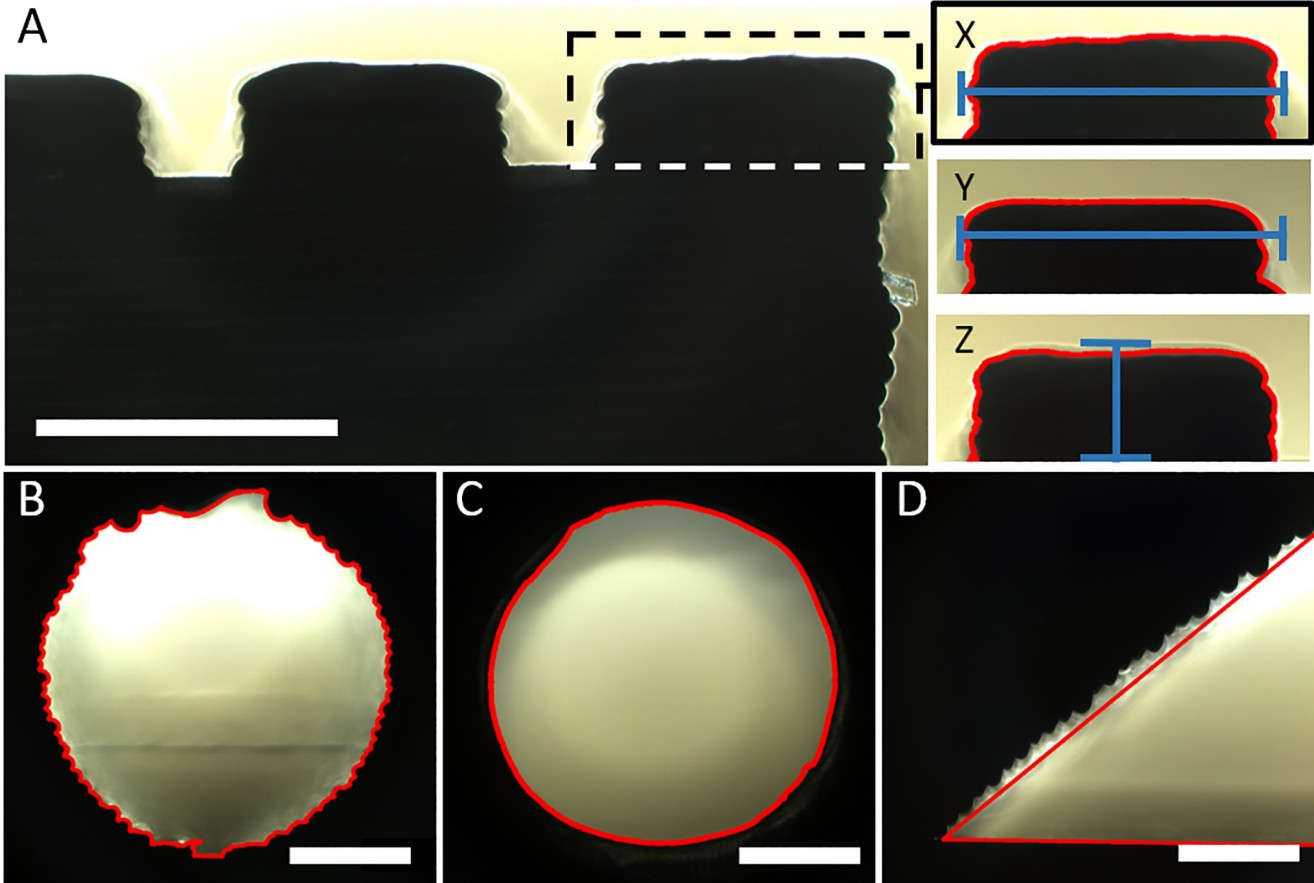

**Fig 3. Evaluation of printed test bodies using Fiji macros.** Microscopic pictures of the test bodies were taken and the area of interest was cropped (dotted line). The shape was identified and analysed using self-written Fiji macros (red line). (A) For X and Y resolution test bodies (X and Y) the maximum width and for the Z resolution test body (Z) the maximum height was measured (blue line). For the horizontal printed channels (B) as well as the vertical printed channels (C), the feret diameter and the roundness was calculated. (D) The angle of the appropriate test body was measured manually using the angle measuring tool from Fiji (red line). Scale bar equals 1000 µm.

linear increase of deviation from 3.6 ± 0.7% at 3000 µm, down to 11.6 ± 0.5% at 1500 µm diameter. Then it had oscillating deviations of -4.0 ± 0.9% (1000 µm), 12.3 ± 0.1% (800 µm), -6.8 ± 0.4% (600 µm) and 12.2 ± 2.0% (400 µm). Roundness of horizontally printed channels revealed an exponential decrease in FFF and SLA (SLS only 2 dimensions measurable). Starting at 0.95 ± 0.01 for FFF and 0.91 ± 0.02 for SLA at 3000 µm diameter values decreased to 0.63 ± 0.06 at 400 µm (FFF) and 0.64 ± 0.06 at 800 µm (SLA). Similar to horizontal channels, SLS was only able to generate channels of 3000 µm diameter in vertical direction, with a deviation of -9.1 ± 0.8%. SLA was able to print channels down to 600 µm diameter, showing again a stable average deviation of 6.1 ± 1.0% from 3000 µm down to 1000 µm diameter. Then accuracy decreased to 2.3 ± 12.3% (800 µm) and 10.0 ± 1.9% (600 µm). According to that FFF, also generated channels down to 600 µm diameter, but showed an exponentially increasing deviation of 0.6 ± 0.4% at 3000 µm to -38.2 ± 6.3% at 600 µm diameter.

In contrast to horizontal channels, the roundness of vertical channels in SLA achieved much lower and stable average deviation of 0.97 ± 0.01 compared to FFF, that roughly showed an exponential tendency of decreasing roundness from 0.96 ± 0.02 (3000 µm) to 0.81 ± 0.05 (600 µm).

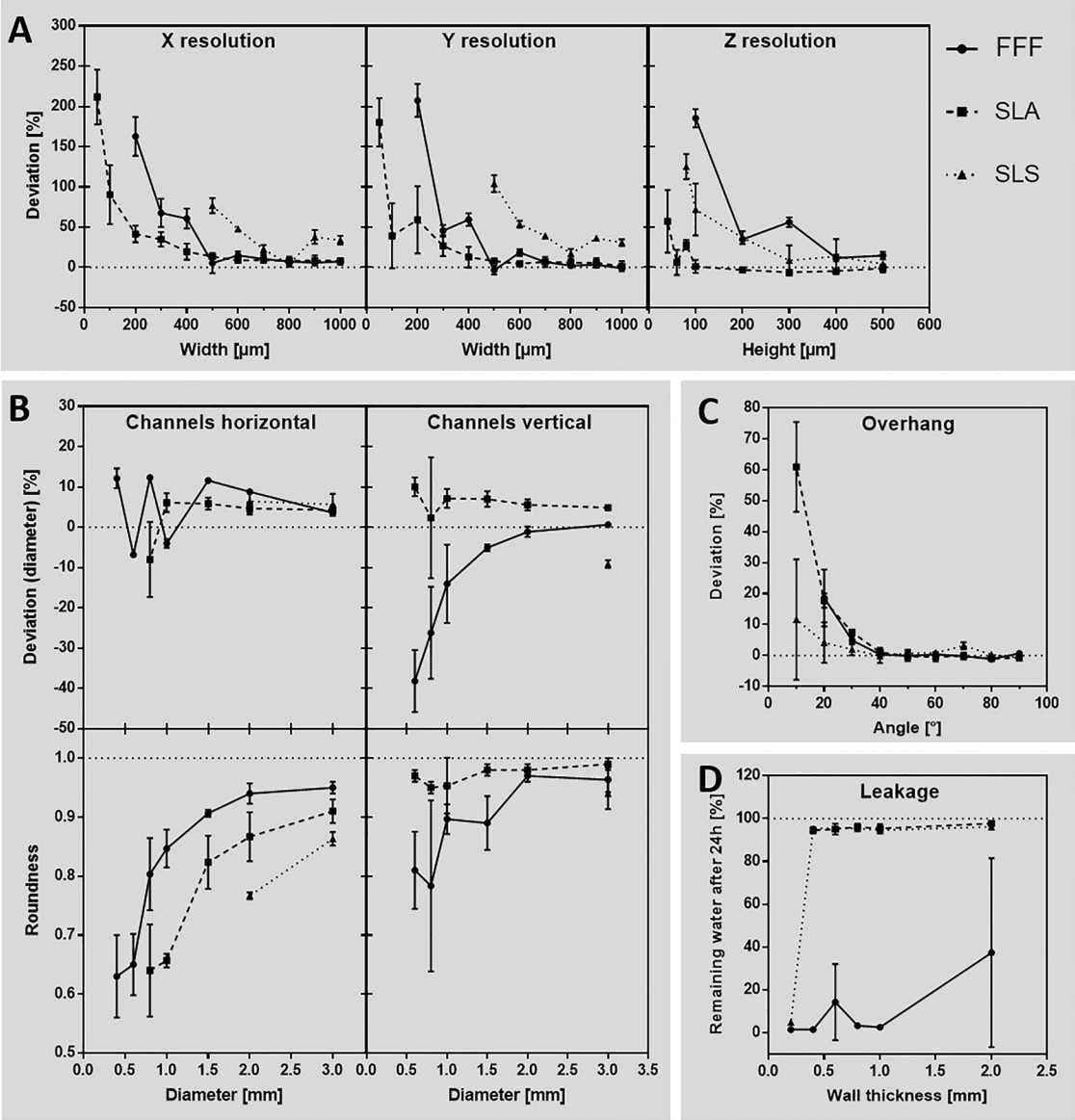

**Fig 4. Accuracy of different 3D printing methods compared to each other.** Test bodies printed with FFF (Raise3D Pro 2 printer), SLA (Form 2 printer) and SLS (Lisa Pro printer) technique were analysed to their deviation of the respective geometry that was designed by CAD before. (A) Accuracy of the base axes X, Y and Z. (B) Diameter and roundness of horizontal and vertical printed channels. (C) Angled overhangs, printed without support structures. (D) Leakage investigated for different wall thicknesses. n = 3.

Printing overhangs without any support structure (Fig 4C) had very low deviations of 0.0 ± 0.6% (FFF), -0.5 ± 0.6% (SLA) and 0.7 ± 1.2% (SLS) from 90° down to 40° in all three printing methods. Angles below 40° showed exponential increasing deviation, while FFF was not able to print 10° overhangs at all. Therefore, 20° overhangs resulted in 18.6 ± 6.6% deviation in FFF. In SLA, 10° were produced with 61.0 ± 11.9% deviation and in SLS 10° deviated by 11.6 ± 15.9%.

Testing the leakage of 3D-printed parts (Fig 4D showed low loss of water or none at all by 95.6 ± 0.9% remaining water after 24 h incubation in cups with 2000 μm down to 400 μm wall

thickness in SLA and SLS. 200 μm wall thickness was not printable in SLA, and showed leakage in SLS (4.9 ± 0.6% remaining water). Although FFF did generate all wall thicknesses from 2000 μm down to 200 μm, only 1 out of 3 samples of 2000 μm and 600 μm wall thickness had no leakage. It is worth mentioning that the accuracy of the printed wall thicknesses are related to the X and Y resolution as described before.

Printing one set of all test bodies simultaneously resulted in printing times of 2.75 h for FFF and SLA, and 6.67 h for SLS.

Since printing accuracy is not the only parameter important for evaluating printing techniques, we also subjectively compared printing time, acquisition and material costs, as well as preparation and post processing effort of the three techniques in relation to each other as shown in Table 2. With our findings in this table we state FFF and SLA as the most reliable printing methods, due to the best price-performance-effort ratio. Even if FFF shows the lowest surface quality, it is the least expensive one and simplest to use. Nevertheless, SLA shows the best absolute printing accuracy and quality of the compared printers and techniques.

## Established guidance system for 3D printing

With our results so far, we derived a guidance system as shown in Fig 5. First, the STL-files of all test bodies as well as mechanical strength bodies (S3 File) are downloaded. Then, the individual printer-software is used to transfer the STL-file into G-code. Within this step, it is crucial to set the printer settings suitable to the desired needs and material. Afterwards the parts are printed. The test bodies are designed in a way that the region of interest can easily be imaged using a microscope. Images are then loaded into Fiji and the area of interest is cropped manually. After that, the Fiji macros, as explained in Fig 3, are executed to quantify the accuracy of the printed parameters. Additionally, the bodies for mechanical tests can be sterilised before investigating properties of different materials and the effect of individual sterilisation methods. For TE approaches the biocompatibility of the bioreactor test body after sterilisation has to be investigated. At last, different printers, materials, printing techniques and printing parameters can be compared and the obtained results can be used to adjust printing parameters for better printing results.

## Optimal printing settings are individual to each device

In order to check the robustness of this guidance, we performed a transregional comparison test of FFF printers with two additional university institutes. For this, we only provided STL-files and filament and no printing parameters were communicated other than printing

**Table 2. Evaluation of the printing techniques to each other.**

|  | FFF | SLA | SLS |
|---|---|---|---|
| **Acquisition costs** | Low | Medium | High |
| **Material costs** | Low | High | Medium |
| **Preparation effort** | Medium | Low | High |
| **Printing time** | Low | Low | High |
| **Post processing effort** | Low | Medium | High |
| **Surface quality** | Low | High | Medium |
| **Accuracy** | Medium | High | Low |
| **Price-performance-effort ratio** | High | High | Low |

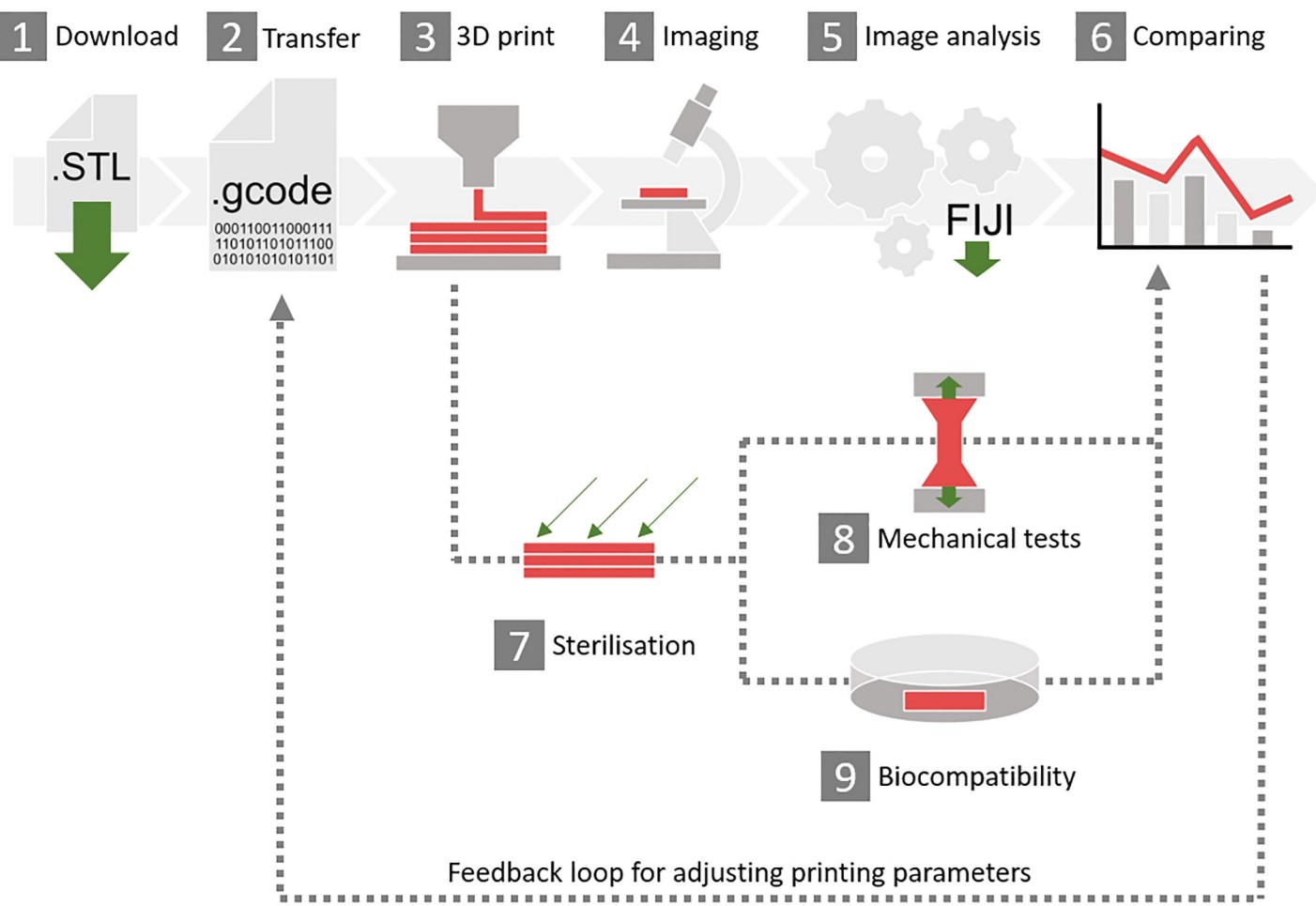

**Fig 5. Overview of the 3D printing guidance.** (1) Download the STL-files of the test bodies attached to this publication. (2) Transfer the STL-files into G-code or similar, using the individual software and printer settings. (3) 3D print the test bodies. (4) Image the printed parts. (5) Download the Fiji macros attached to this publication and analyse the images after cropping the area of interest. (6) Compare the results to each other or to different printers if they match the desired quality or not. After printing, the sterilisation test (7), mechanical test (EN ISO 527 and 178) (8) or biocompatibility tests (ISO 10993–5) (9) can be performed.

temperature and print bed temperature since these are material related. In this case, Raise3D Pro 2 (RP2) (RAISE3D Technologies, USA), Ultimaker 3 (UM3) and Ultimaker 5S (UM5) (Ultimaker B.V., Netherlands) were compared (Fig 6). Data of the RP2 printer was already used as a FFF reference before, so the shown data here for this printer is identical to the data shown in Fig 4. RP2 achieved a stable deviation of 6.5 ± 7.4% down to 500 μm when analysing X and Y resolution (Fig 6A) followed by an exponential increase of inaccuracy of 185.1 ± 28.9% at 200 μm lowest possible dimension. UM3 and UM5 did not print dimensions smaller than 400 μm and showed exponentially increasing inaccuracy from 28.4 ± 7.3% at 1000 μm to 100.5 ± 16.5% at 400 μm. In contrast, accuracy deviation in height was low in UM3 and UM5 down to 200 μm (16.7 ± 5.5%), followed by deviating 635.7 ± 14% from 20 μm height in UM3. UM5 did stop printing heights below 60 μm (147.8 ± 21.5%) and RP2 below 100 μm (185.3 ± 9.3).

As shown before, RP2 had oscillating deviation in horizontal printed channels. UM3 and UM5 both result in a smaller diameter than designed initially (Fig 6B). The printing limit of all

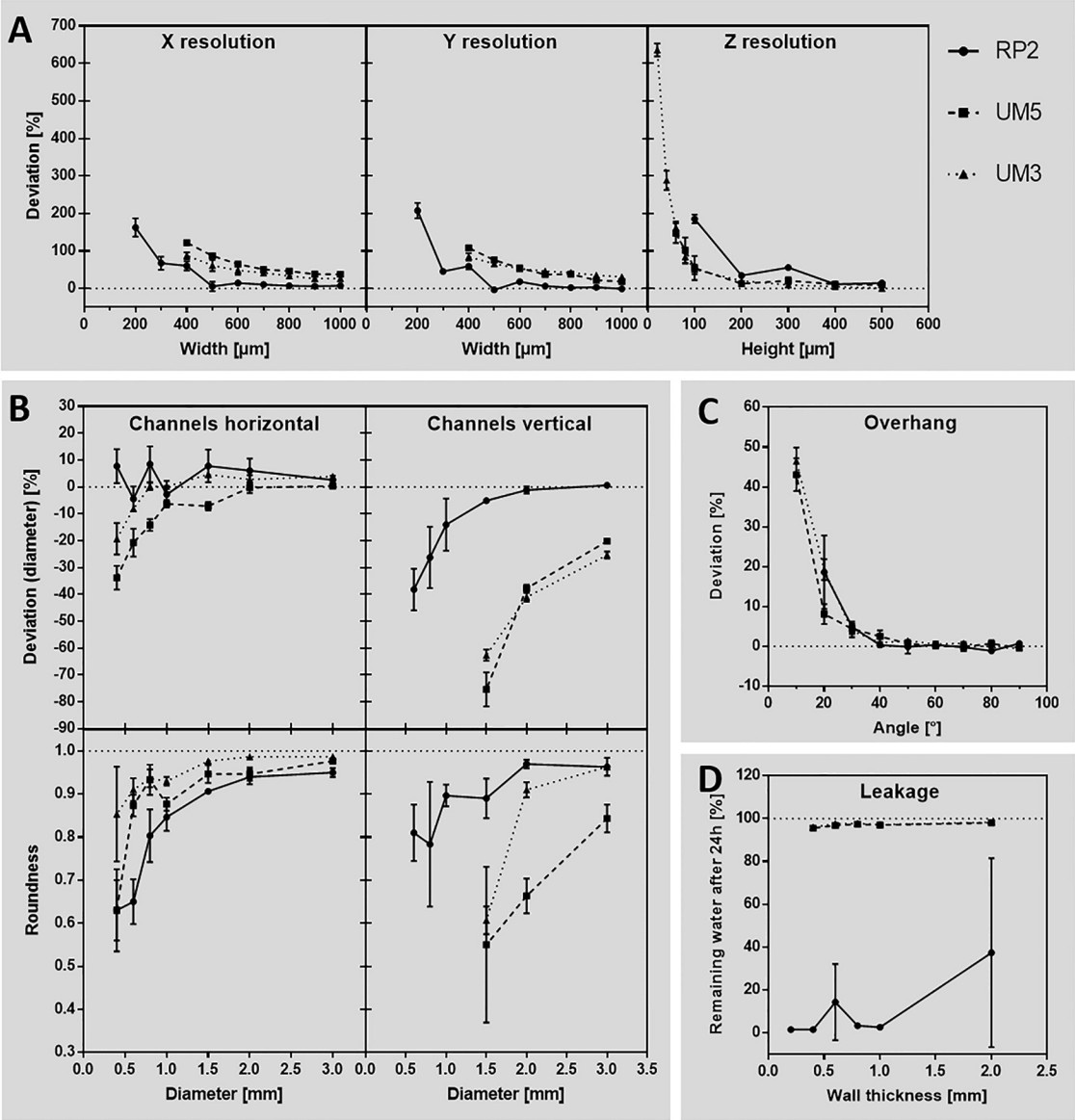

**Fig 6. Accuracy of different FFF printers compared to each other.** Test bodies, which were FFF-printed with Raise3D Pro 2 (RP2), Ultimaker 3 (UM3) and Ultimaker S5 printer (UM5) were analysed regarding their deviation of the respective geometry that was designed by CAD before. Data from Raise3D Pro 2 are equal to Fig 4 (FFF). (A) Accuracy of the base axes X, Y and Z. (B) Diameter and roundness of horizontal and vertical printed channels. (C) Angled overhangs, printed without support structures. (D) Leakage investigated for different wall thicknesses. n = 3.

printers was 400 μm diameter channels. Roundness was decreasing more in RP2 (0.63 ± 0.06) and UM5 (0.63 ± 0.08) compared to UM3 (0.85 ± 0.09) at 400 μm. Loss of quality also showed an exponential tendency here. In printing vertical channels, the ultimaker printers achieved significantly different capabilities than RP2. Deviation was already at -22.8 ± 2.8% at 3000 μm diameter and it further decreased to -69.0 ± 7.5% at 1500 μm diameter which was also the lowest diameter printable. According to that, roundness also decreased rapidly from 0.90 ± 0.06 (3000 μm) to 0.58 ± 0.11 (1500 μm). By printing overhangs (Fig 6C), all printers showed a 0.4 ± 1.0% deviation down to 40°, following by exponential increase to 43.0 ± 3.4% (UM5) and

46.4 ± 2.8% (UM3) at 10˚ and 18.6 ± 7.5% (RP2) at 20˚. In contrast to RP2, UM3 and UM5 did not show any leakage in all wall thicknesses with an average amount of remaining water of 97.2 ± 0.8% for both printers (Fig 6D). The wall thickness of 200 μm was only printable with the RP2 printer. As already shown before, printing time was about 2.75 h for the RP2 printer. UM3 and UM5 show an average printing time of a whole set of test bodies of respectively 4.83 h and 2.83 h. These results show a successful application of our established guidance in the described interlaboratory tests and thereby confirmed its robustness and reproducibility.

## Multiple autoclaving corrupts material properties

Prior to usage of bioreactors in contact with living cells, any printed part has to be sterilised. Sterilisation methods might affect the materials properties, its interaction with cells as well as the bioreactor lifetime. Here, we compared two different sterilisation processes: Autoclaving, which represents the gold standard in biological research, and vaporised hydrogen peroxide plasma sterilisation, which is much more gentle to the used material. Test bodies were autoclaved and plasma sterilised up to 3 times and material properties such as the tensile and flexural strength were analysed using tension and bending tests according to EN ISO 527 and 178 (Fig 7).

FFF-printed parts made from lignin-based Green-TEC Pro filament (Extrudr | FD3D GmbH, Austria) showed an untreated tensile strength of 44.4 ± 9.2 $N/mm^2$ and a flexural strength of 42.7 ± 3.3 $N/mm^2$. After performing 3 times plasma sterilisation, no significant difference was detected in both tensile and flexural strength. In contrast, autoclaving significantly reduced tensile (14.8 ± 3.6 $N/mm^2$) and flexural strength (6.2 ± 1.1 $N/mm^2$) after 3 treatments. Flexural strength was already decreased significantly after the first autoclaving treatment. Untreated parts, printed from bisphenol A ethoxylate-based Dental SG Resign (Formlabs Inc., USA) by SLA technique showed a tensile strength of 30.7 ± 7.9 $N/mm^2$ and a flexural strength of 27.0 ± 6.3 $N/mm^2$. Autoclaving and plasma sterilisation resulted in no significant impact on the material after 3 treatments. Autoclaving however tends to reduce tensile (14.0 ± 4.3 $N/mm^2$) and flexural strengths (20.4 ± 2.2 $N/mm^2$). Untreated SLS printed parts made from Nylon 12 (Sinterit sp. z o.o., Poland) had a tensile strength of 26.5 ± 0.7 $N/mm^2$ and a flexural strength of 26.8 ± 6.2 $N/mm^2$. After the first treatment of autoclaving and plasma sterilisation, an increase in tensile strength was measured. However, after the second and third treatments in both sterilisation processes no significant differences were measured in the value of the tensile strength in comparison to the untreated control.

## 3D printed bioreactor allows culture of a perfused collagen hydrogel

To verify the applicability of 3D printing in TE after selection of a suitable printing method and identification of the printer limitations and tolerances, a sophisticated bioreactor was designed to culture a collagen hydrogel under dynamic flow conditions as proof of concept. Therefore, a complex multi-piece bioreactor was SLA-printed using Dental SG Resin. The design enables the generation and perfusion of a central channel in the hydrogel to supply nutrients to the tissue and to apply shear stress for proper maturation of endothelial cells. Furthermore, barb-like structures at the in- and outlet of the channel support anchoring the collagen hydrogel to it and thus preventing the detachment of the hydrogel. Still, the bioreactor interior shows a round base area to facilitate uniform contraction of the collagen gel if necessary. Moreover, the design of the bioreactor is customised for easy handling, e.g. assembling and filling, for adequate supply and sustainable use of materials. In addition, the bioreactor has not to leak, biocompatible, reusable and easy to clean.

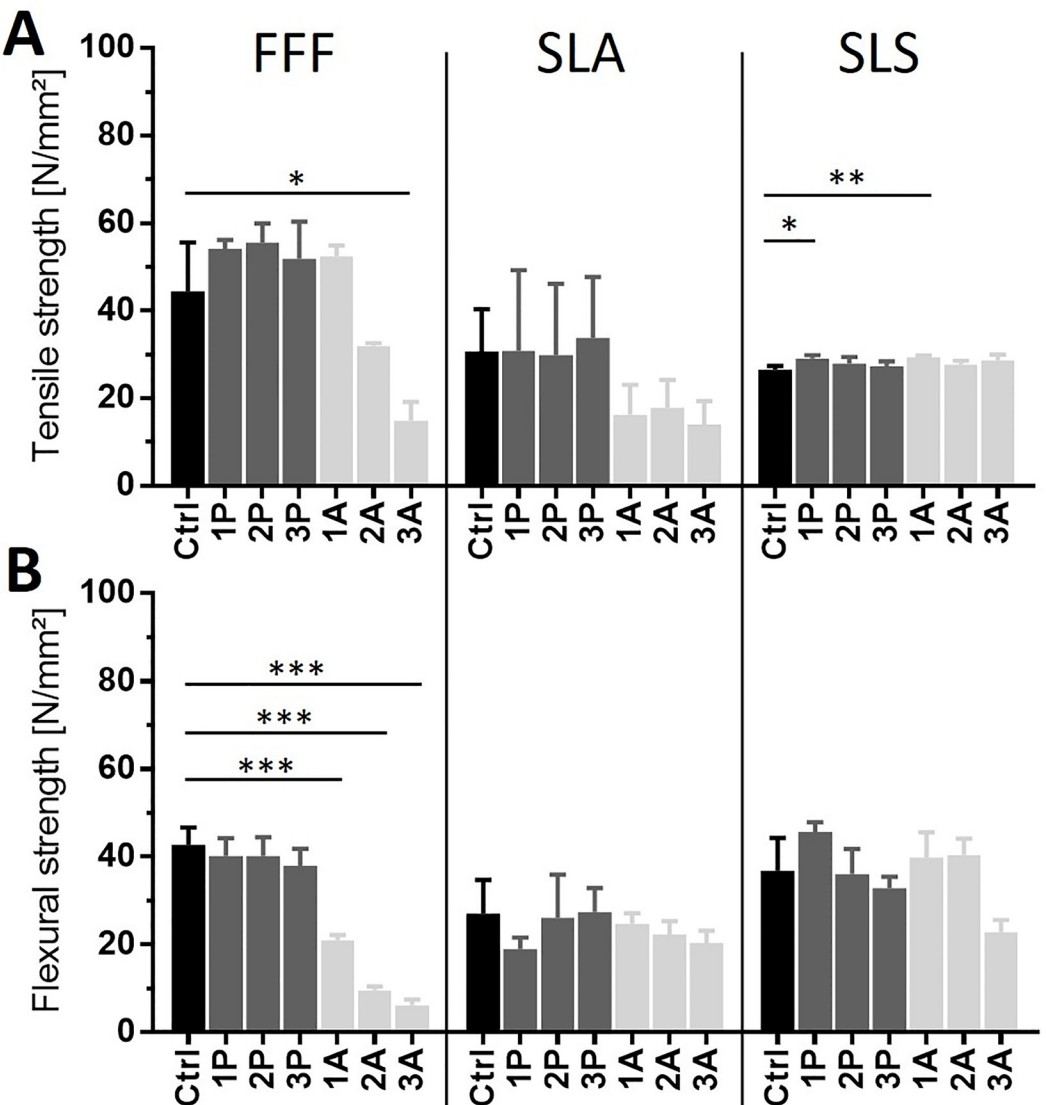

**Fig 7. Effect of autoclaving and plasma sterilisation to printed materials.** Tension (A) and bending (B) test were performed according to EN ISO 527 and 178. FFF printed material showed no change in material properties when treated by plasma sterilisation up to 3 times, but autoclaving significantly reduced tensile and flexural strength of the material. In SLA-printed material, no significant loss of tensile and flexural strength could be measured, but autoclaving tends to be more corruptive to the material over time. Materials printed by SLS show a significant increase of the tensile strength when sterilised for the first time, but no further impact when sterilised up to 3 times. In flexural strength, however, both methods tend to downgrade the material properties but do not show significances. All significances are referring to the control and are indicated by *. n = 3.

Due to the resulting complex geometry, it would be more difficult and expensive to apply classical manufacturing methods. To verify this statement, we designed a slightly altered bioreactor: In order to be milled, the bioreactor body had to be divided into several pieces (Fig 8). The milled bioreactor was composed of 5 parts. Additionally, two barb-like structures as well as two connectors for the peripheral venous catheters had to be screwed into the in- and outlet of the bioreactor. Summarising, the milled bioreactor would consist of 9 parts as opposed to 3 parts for the 3D printed bioreactor. Given that the whole bioreactor must be tightly sealed,

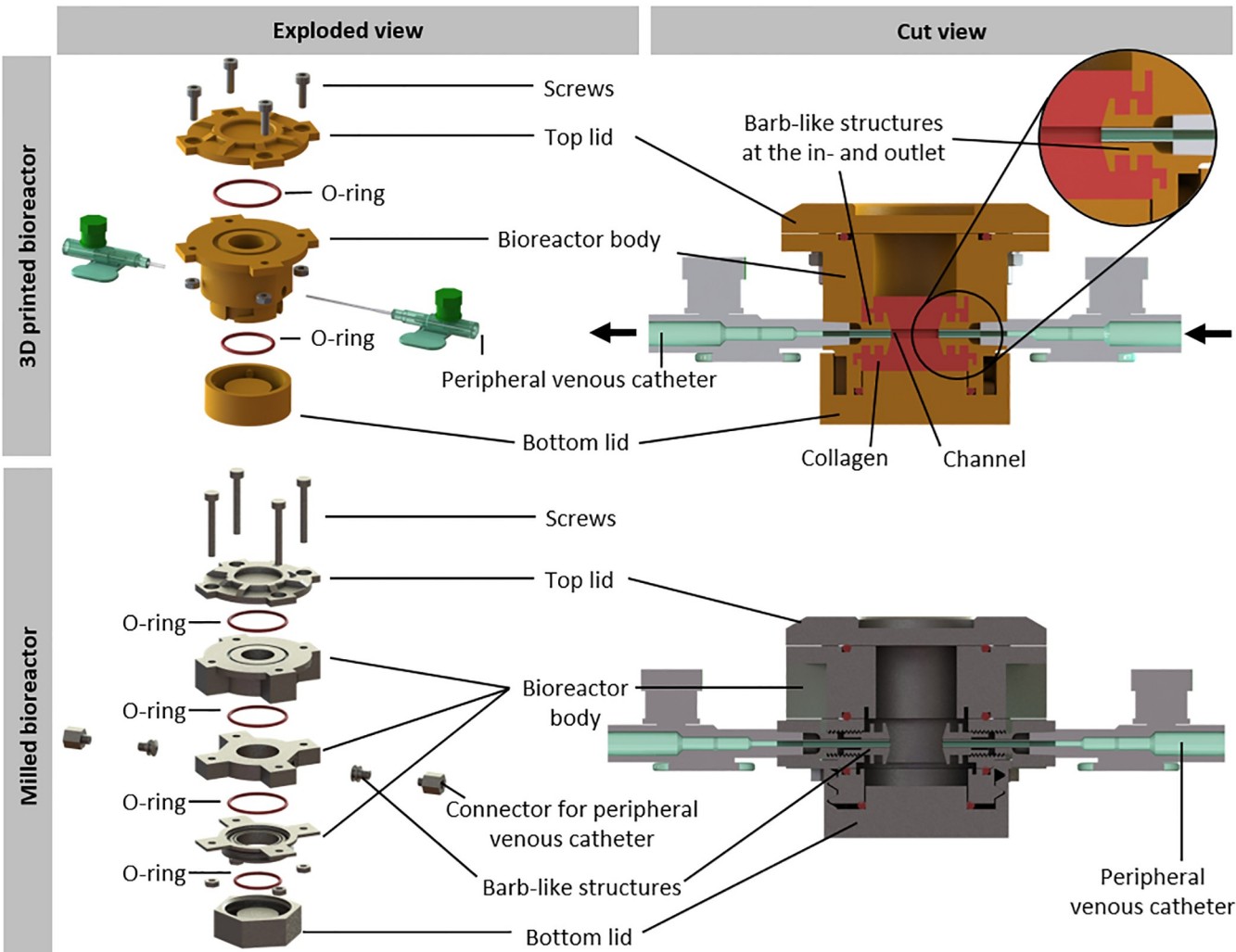

**Fig 8. Comparison of 3D printed and milled bioreactor system for the culture of perfused hydrogels.** Overview of the 3D printed and the milled bioreactor. Both bioreactors are closed with a top lid and a bottom lid, sealed via O-rings. The bottom is connected to the bioreactor via a plug connection, whereas the top lid is fixed with screws. In contrast to the 3D printed bioreactor, the milled bioreactor consists of nine parts. The bioreactor body has two opposing inlets to hold two peripheral venous catheters for perfusion. Barb-like structures protrude inwards from the openings to anchor the hydrogel. For the 3D printed bioreactor the barb-like structures are part of the bioreactor body. For the milled bioreactor they are screwed into the openings as well as the connectors for the peripheral venous catheters. Arrows indicate the medium flow.

every additional part increases the risk of leakage. Moreover, having less parts allows easier handling and higher robustness.

Another aspect was the cost of manufacturing the bioreactor which is for the 3D printed one about 76 ± 23 € and for the milled one, made from stainless steel, about 1590 ± 496 € (3 quotations from independent companies), showing milling is about 21-fold more expensive than 3D printing. Certainly, a reactor milled from stainless steel can be used more often, but, as it was mentioned before, using 3D printing can produce the same bioreactor with less number of pieces at a lower price and developmental changes can be adapted quickly. Therefore, the classical manufacturing method is not economically reasonable.

The tailored 3D printed bioreactor provides a suitable environment for engineered vascularised tissue. Therefore, it mainly consists of the bioreactor body, the bottom and the lid

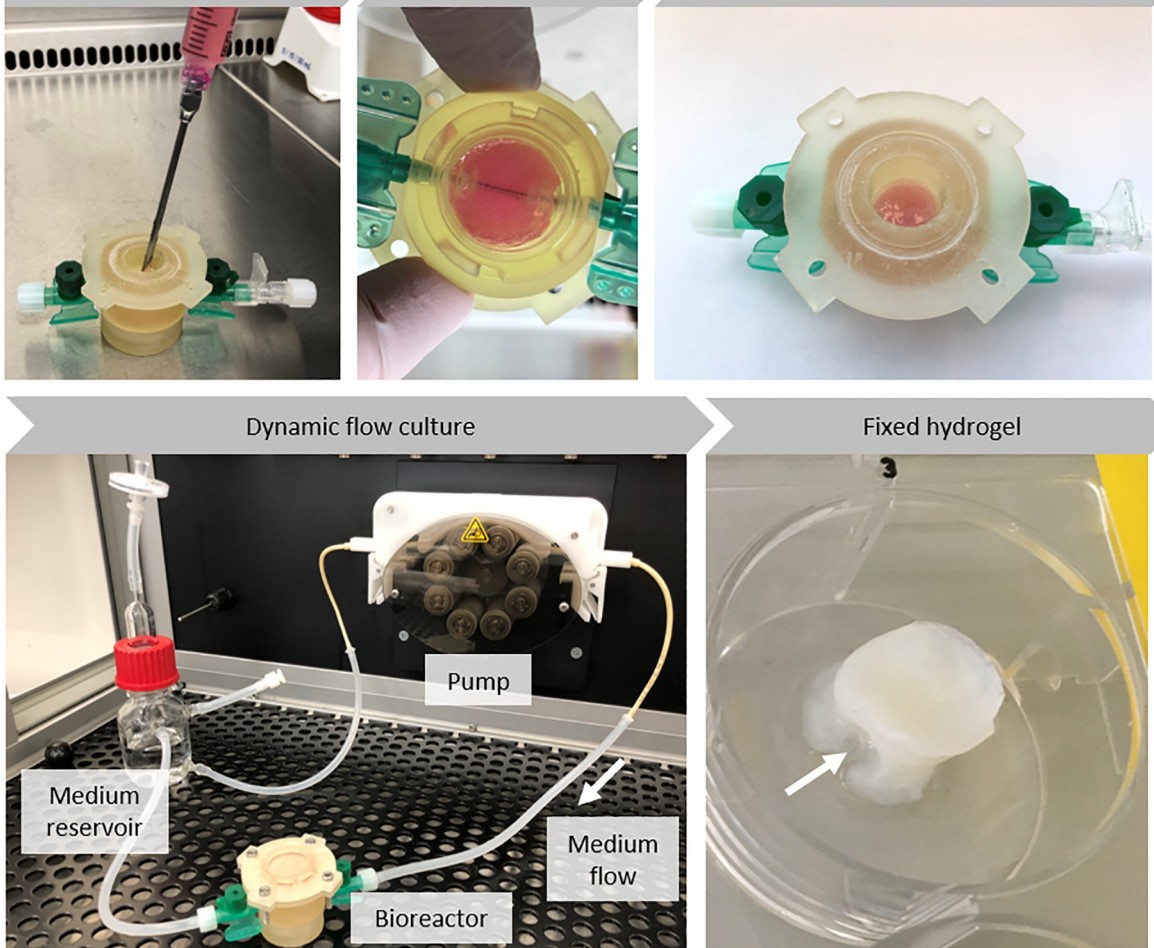

**Fig 9. Workflow for starting the bioreactor with cell-laden collagen hydrogel.** HdF-containing collagen gel is cast in the assembled bioreactor enclosing the inserted cannula of the peripheral venous catheter. After 24 h, the cannula is removed. Thereby, a channel-like structure is created, which enables perfusion of the hydrogel with medium. HdmECs are seeded into the channel and the hydrogel is cultured under dynamic flow conditions. After 14 days, the hydrogel is removed from the bioreactor for further analysis. The arrow shows the location of the channel in the fixed hydrogel.

(Fig 8). O-rings ensure a tight sealing. The bottom is fixed to the bioreactor by a plug connection, whereas the lid is closed by screws. To form a central channel for perfusion, two peripheral venous catheters are inserted through the in- and outlet of the bioreactor body. During assembly of the bioreactor, the cannula of one peripheral venous catheter is inserted through the in- and outlet (Fig 8). Collagen gel is cast around the cannula, which is removed after the hydrogel has polymerised, generating a channel-like structure in the tissue construct (Fig 9). To prevent detaching of the collagen hydrogel from the bioreactor during maturation, barb-like structures are printed at the in- and outlet. Here, the Luer taper connectors of the peripheral venous catheters connect the bioreactor to the tubing system of the circuit medium flow for perfusion culture.

The workflow for starting the bioreactor with biofabricated, cell-laden hydrogel is depicted in Fig 9: The assembled and sterilised bioreactor was filled with hdF containing collagen hydrogel enclosing the cannula. After polymerisation, the cannula was removed, thereby creating a channel-like structure through the hydrogel and enabling perfusing with cell culture

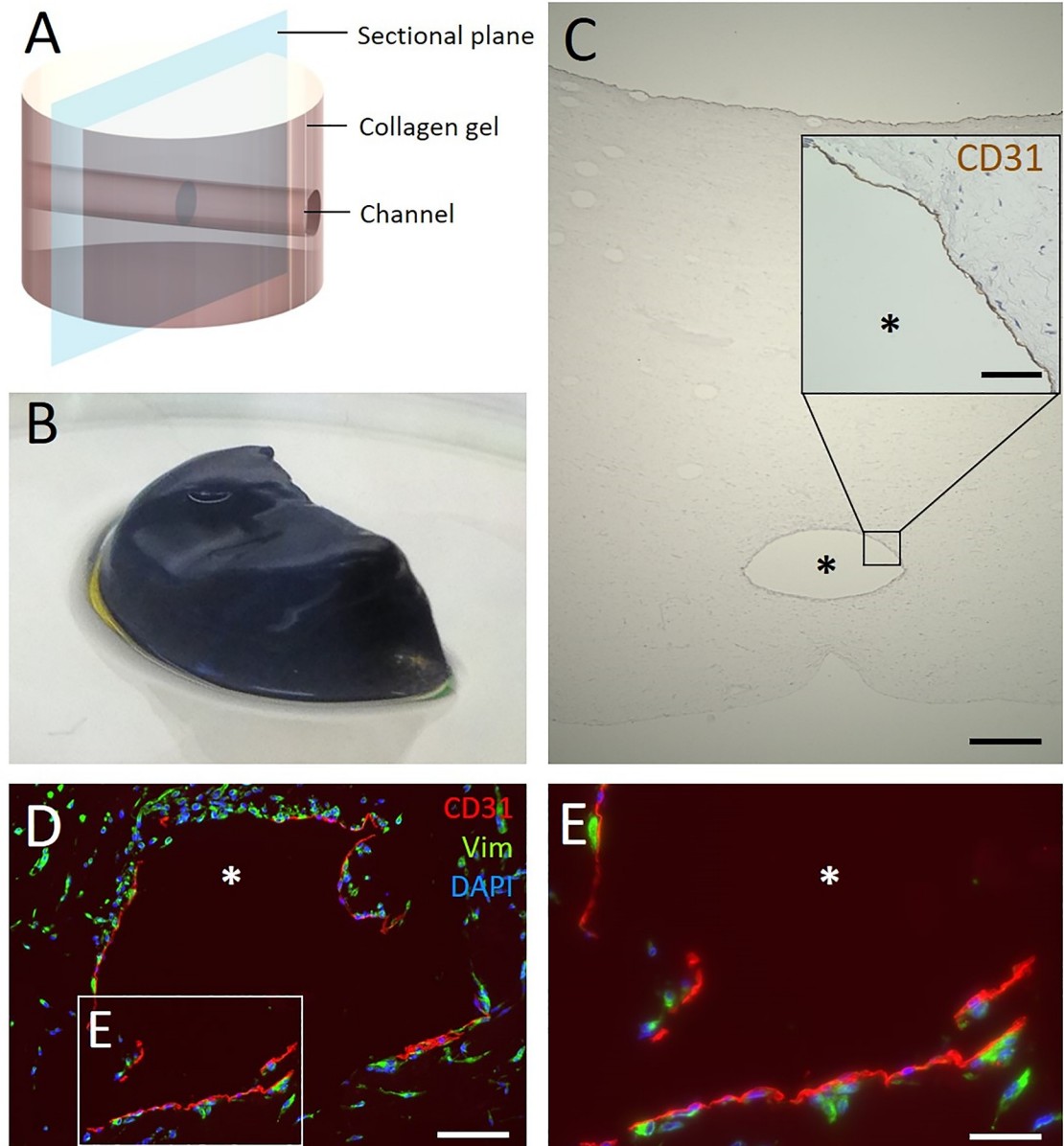

**Fig 10. Analysis of the biofabricated hydrogel with channel.** (A) The schematic representation of the collagen hydrogel shows the position of the channels and the sectional plane for the following cross-sections. (B) Qualitative MTT-staining of half of a tissue construct. The blue color indicates viable cells within the hydrogel. (C) The central channel is visible in the HE-stained cross-section of the hydrogel after 14 days of culture in the bioreactor under dynamic flow conditions. The inset shows in detail the immunohistological staining of the channel. It reveals CD31-positive cells lining the channel lumen, indicating colonisation with endothelial cells. (D) Immunofluorescence staining of cross-sections of the channel visualizes the presence of endothelial cells and fibroblasts. Positive staining for CD31 (red) shows endothelial cells at the channel surface and positive staining for vimentin (green) shows fibroblasts within the hydrogel. Cell nuclei are labeled with DAPI (blue). (E) Magnification of D. The asterisk marks the channel lumen. Scale bars 500 μm (C), 100 μm (D) and 50 μm (inset in C, E).

medium. The next day, hdmECs were seeded into the channel. To allow for adherence, the hydrogel was statically cultured for 4 h at 37°C before reconnecting the bioreactor to the fluidic circuit. The cell-laden hydrogel was cultured under dynamic flow conditions for up to two weeks and finally removed from the bioreactor and fixed for subsequent analysis. Analysis included a qualitative MTT assay to evaluate the effectiveness of perfusion (Fig 10B). Blue

staining indicated metabolic active cells, thereby proving viability of the cells within the collagen hydrogel during the two weeks of culture.

Fig 10A depicts a scheme of the tissue construct with the channel-like structure and visualises the sectional plane for the following stainings. First, Fig 10C shows an overview of a HE-stained cross-section of the hydrogel. After 14 days of culture in the bioreactor system, the created channel was visible in the hydrogel and the surrounding collagen gel was smooth with some trapped minor air bubbles. The hdmEC colonised the channel by lining the lumen surface, indicated by an immunohistological staining for the endothelial cell marker CD31 (Fig 10C). Immunofluorescence staining shows homogeneously distributed vimentin-positive cells, demonstrating the presence of hdF within the hydrogel. CD31-positive cells were only located at the channel lumen forming a monolayer (Fig 10E).

## Discussion

Constructing parts by 3D printing is widely used in different industrial fields and also used in life science nowadays [39,40]. The appreciable advantage is the capability to generate complex geometries by simply designing a part virtually and then using a 3D printer to make the part just like one would print a text file in 2D. Thus, engineering know-how and lengthy production times can be avoided. However, additive manufacturing is complex in reality, requires special know-how and has therefore rarely been implemented in TE. Since 3D printing allows a high degree of freedom in designing the geometry, choice of material, printing technology, and printing parameters, manufacturing a tailored bioreactor additively comes with a lot of trial and error studies resulting in time-intensive iterations and waste of resources. Especially for research groups without access to engineering and material knowledge, the field of 3D printing can be challenging. Therefore, we systematically assessed common techniques and materials to give non-experts a guidance for getting started and using 3D printers for TE, as well as reducing iteration steps.

First, we designed a set of test bodies to investigate the printers limitations. Unlike already existing test bodies, ours are optimised for microscopic analysis and, since every parameter has its own test body, the individual parameters are not dependent of each other. These test bodies are suitable for assessing different 3D printing methods, as we could show by comparing printers representing the three major printing techniques FFF, SLA and SLS. Within this comparative study part, we were able to evaluate their advantages and disadvantages related to each other. For example, FFF has proven to be the technique that requires the least investment. Also, it has the lowest post processing effort and shares the shortest printing time together with the SLA technique. These advantages make the FFF technique the most suitable for beginners and low budget solutions, even if the surface quality and the printing accuracy are rather low, compared to the other printing methods. Surface quality in FFF printing is highly influenced by the barus-effect, which we believe, has a major impact on the deviation in accuracy. The barus-effect or die swell states the swelling behavior of viscous polymer solutions forced through narrow openings. In contrast, SLA printing demonstrated the best surface quality and printing accuracy of all tested techniques, as well as low printing times and preparation effort, which makes it the best technique for printing high complex and detailed geometries. Nevertheless, device and material costs are much higher in SLA printing compared to FFF. In our experimental setup, SLS printing was not reliable. Besides being the most expensive printer, it also required the most preparation and post processing effort as well as high printing times and showed poor surface quality as well. SLS printing might have its advantages in different industrial settings, like metal 3D printing, but our results indicate that it is not suitable for TE approaches like the one we described here. The parameters analysed in this work are only a

small representation of all the modifiable parameters in 3D printing. For example, in SLA we were able to generate even more complex geometries by printing it tilted by 45˚, while less leakage could be achieved with higher infill densities and a higher number of shells. Every parameter in each distinct device had a very special effect on the resulting printing time and quality, which is the reason why no global 3D printing parameters can be defined for optimal results. Hence, single parameters can not be compared equivalently between different devices, therefore everyone has to test this for his own needs. Thus, it is important for our test bodies to be printable independently in different labs. That was proven by testing the robustness of our guidance on the example of multiple FFF printers.

Due to their rather cheap production costs, 3D printed parts are usually stated as single-use components. Since the focus in TE approaches is on high complex composite materials with a variety of physical properties, it is not sustainable for every printable material to handle it as single-use. This might be neglectable in most materials available today, but since 3D printable materials are getting more complex, the importance of sustainability will increase in future. Due to the multiple use of printed parts for biological experiments, it is crucial that the material can be sterilised several times. As sterilisation processes might affect the material properties, we systematically analysed the flexural and tensile strength of the material after repeated sterilisation with different procedures. As an alternative to the autoclaving gold standard, the vaporised hydrogen peroxide plasma sterilisation as a more gentle sterilisation process was used. It has been shown that the impact of the sterilisation methods highly depends on the material itself. In FFF printing (Lignin-based material) autoclaving results in significant loss of flexural and tensile strength. In SLA (Bisphenol A ethoxylate-based material), these impacts are not significant but still show the same tendency as in FFF. In SLS (Nylon 12-based material) however, both sterilisation methods showed the same impact. It can be expected that more than 3 treatments will further increase the effect of material corruption. We showed that the material properties are no longer the same as the original ones after various sterilisation treatments, which must be taken into account when using the material for TE approaches. Additional sterilisation methods like alcoholic disinfection, UV or gamma radiation and other treatments are possible as well.

As proof of concept and translation of our findings, we used additive manufacturing to produce a complex bioreactor for an engineered tissue construct. To decide which printing method to use, we applied our guidance system (Fig 5). Both, FFF and SLA showed the best price-performance-effort ratio as depicted in Table 2. As surface quality and accuracy were more important for our application, we chose SLA technique as most suitable for printing the bioreactor. A crucial problem is the discrepancy between the conceptualised and the printed part. Accordingly to the gained knowledge about tolerances and limitations of the printer, the designed geometry of the bioreactor was adapted. Vascularisation is one of the key challenges in TE and requires more sophisticated bioreactor technology. Therefore, we designed a bioreactor for a perfused collagen hydrogel to provide a base to develop somatic tissue constructs including a connective tissue component that can be perfused through a tubular structure lined with endothelial cells. The perfusion enables the nutrition of the collagen gel and preferably promotes vessel maturation via shear stress. To meet the needs of this vascularised scaffold, a bioreactor was designed that was mainly composed of a bottom lid, the bioreactor body and a top lid (Fig 8). As cell-laden collagen gels tend to shrink due to remodelling by the fibroblasts, barb-like structures were used to anchor the gel to the connectors and thereby prevent detaching and loss of perfusion [56]. Due to its complex geometry, producing the bioreactor by 3D printing is favourable compared to classical manufacturing methods like milling. The latter would require more parts, which have to be connected tightly and therefore bear the risk of leakage. Additionally, milling is 21-fold more expensive than 3D printing, making the latter

economically unbeatable. In-house 3D printing of bioreactors facilitates the development of prototypes and therefore contributes to faster scientific progress.

The 3D printed bioreactor was used to culture a cell-laden hydrogel perfused by a central channel seeded with endothelial cells. Metabolic activity of the cells within the hydrogel was assessed via a MTT assay, showing the viability of the tissue construct. Histological analysis showed the existence of the channel and lining of the channel lumen with endothelial cells (Fig 10). These are first crucial steps towards perfused vessels and later on vascular networks.

Concluding these results, we proved the possibility to produce a complex bioreactor for a perfused TE construct by 3D printing after identifying a suitable printing method by applying our established guidance. Besides being advantageous for proper maturation of vessel-forming endothelial cells, the perfusion allows the culture of larger tissue constructs and enables therapeutic testing like systemic administration of drugs or analyses of penetrated substances. Thus, the tissue construct is versatile for a wide range of research purposes such as penetration studies, evaluation of proper treatment in personalised medicine or investigation on metastasis development. Depending on cells, scaffolds and individual design, different tissue constructs, like vascularised skin equivalents or else, could be generated this way. Additionally, this is a great achievement in terms of sustainability like production time, transport, resource consumption and waste generation.

## Conclusion

We established a guidance system for systematically optimising 3D printing for TE approaches. As proof of concept, we used this guidance system to determine a suitable printer for production of a 3D printed complex bioreactor that also enables tissue perfusion. Since the manufacturing of this bioreactor would have been much more resource intensive if produced by commercial subtractive methods, 3D printing showed to be significantly more sustainable. With this printed bioreactor we were able to generate a perfused, collagen-based tissue construct containing primary human fibroblasts as well as human microvascular endothelial cells lining the channel lumen. Hence, using this guidance is an easy way for tissue engineering groups to enter 3D printing systematically and thereby boosting individual additive manufacturing in life science.

## Supporting information

**S1 Fig. Code for Fiji macros.** The resulting geometries of the printed test bodies were visualised by microscopic pictures and then analysed by the Fiji macros shown here. For quantification of the shape of the basic axes test bodies X, Y and Z the same macro was used (Macro XYZ). For quantification of the channels diameter and roundness a second macro was used (Macro channels). The area of interest was cropped before applying the macros. In both macros, the picture was set to 8-bit grayscale first. Following, an unsharp mask filter was applied to increase contrast. This filter was set to radius 100 (Macro XYZ), respectively 10 (Macro channels) and mask 0.9 for both. Then a grey level threshold (RenyiEntropy), set to 0/70 for Macro XYZ and 40/225 for Macro channels, was applied to reduce noise. Finally, the particle analyser tool identified the shape of the area of interest. Particle analyser was set to size 1000/infinity in both macros. Additionally, the circularity limit was set to 0.2/1 to ignore remaining artifacts in Macro channels.
(TIF)

**S1 File. Fiji code.** IJM-files of Fiji macros ready to use in Fiji.
(ZIP)

**S2 File. All test bodies.** STL-file with all test bodies arranged.
(STL)

**S3 File. Single test body collection.** All test bodies as single STL-files.
(ZIP)

## Acknowledgments

We thank Tobias Schmitz for supporting the plasma sterilisation process and Philipp Fey for photographing the printed test bodies and his support with writing the Fiji macros. Many thanks to Tesda Barthel for the linguistic revision of the draft.

## Author Contributions

**Conceptualization:** Florian Groeber-Becker, Jan Hansmann.

**Formal analysis:** Marius Gensler.

**Funding acquisition:** Florian Groeber-Becker, Jan Hansmann.

**Investigation:** Marius Gensler, Anna Leikeim, Marc Möllmann, Miriam Komma, Susanne Heid, Claudia Müller.

**Methodology:** Marius Gensler, Anna Leikeim, Marc Möllmann, Miriam Komma.

**Project administration:** Florian Groeber-Becker, Jan Hansmann.

**Visualization:** Marius Gensler, Anna Leikeim.

**Writing – original draft:** Marius Gensler, Anna Leikeim.

**Writing – review & editing:** Aldo R. Boccaccini, Sahar Salehi, Florian Groeber-Becker, Jan Hansmann.

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
