## [Decision Letter · Decision Letter 0]

21 Sep 2020

PONE-D-20-27307

3D printing of bioreactors in tissue engineering: A generalised approach

PLOS ONE

Dear Dr. Gensler,

Thank you for submitting your manuscript to PLOS ONE. After careful consideration, we feel that it has merit but does not fully meet PLOS ONE’s publication criteria as it currently stands. Therefore, we invite you to submit a revised version of the manuscript that addresses the points raised during the review process.

Please update your figures, references and writing. For more details, please refer to reviewer 2.

We look forward to receiving your revised manuscript.

Kind regards,

Yi Cao

Academic Editor

PLOS ONE

Journal Requirements:

Reviewers' comments:

Reviewer's Responses to Questions

**Comments to the Author**

1. Is the manuscript technically sound, and do the data support the conclusions?

Reviewer #1: Yes

Reviewer #2: Yes

2. Has the statistical analysis been performed appropriately and rigorously? 

Reviewer #1: Yes

Reviewer #2: Yes

3. Have the authors made all data underlying the findings in their manuscript fully available?

Reviewer #1: Yes

Reviewer #2: Yes

4. Is the manuscript presented in an intelligible fashion and written in standard English?

Reviewer #1: Yes

Reviewer #2: Yes

5. Review Comments to the Author

Reviewer #1: In this manuscript, the authors tried to report a general approach to prepare the bioreactors via 3D printing for tissue engineering applications, not typical subtractive manufacturing methods. These work offer some summary and guideline information for details such as accuracy, cost, price-performance-effort ratio, post processing, product geometries, specific capabilities, limits of various additive manufacturing methods, which is also proofed and practiced by the corresponding experimental results via this guidance system.

Overall, this is a very extensive and well-organized study, which is also helpful to deeply identify a more suitable method for the manufacturing of a complex bioreactor system, I think it shoud be acceptable and I'd like recommend this work to be published in this journal.

Reviewer #2: The manuscript entitled “3D printing of bioreactors in tissue engineering: A generalised approach” was designed a guidance including test bodies to elucidate the real printing performance for a given printer system. In this manuscript, the authors have done a lot of very meaningful work, performance parameters such as the accuracy and mechanical stability of the test body are analyzed, and post-processing steps such as sterilization or cleaning are also considered. The specific comments as follows:

1. What problems should be paid attention to in the design of bioreactor?

2. Please explain the difference and connection between bioreactor and bionic.

3. As we all know, 3D printing still has many problems, including high cost, limited materials, insufficient precision, etc. How to consider and solve these problems in this article?

4. All figures need to be modified for higher quality.

5. The quality of English needs improving. The manuscript should be concise and emphasize the most important points of your work.

6. Most of the references are relatively old, I recommend the authors to cite new references in recent years.

6. PLOS authors have the option to publish the peer review history of their article (what does this mean?). If published, this will include your full peer review and any attached files.

Reviewer #1: No

Reviewer #2: **Yes: **Zhipeng Gu

---

## [Author Response · Author response to Decision Letter 0]

3 Nov 2020

Reviewer #1:

In this manuscript, the authors tried to report a general approach to prepare the bioreactors via 3D printing for tissue engineering applications, not typical subtractive manufacturing methods. These work offer some summary and guideline information for details such as accuracy, cost, price-performance-effort ratio, post processing, product geometries, specific capabilities, limits of various additive manufacturing methods, which is also proofed and practiced by the corresponding experimental results via this guidance system.

Overall, this is a very extensive and well-organized study, which is also helpful to deeply identify a more suitable method for the manufacturing of a complex bioreactor system, I think it shoud be acceptable and I'd like recommend this work to be published in this journal.

Response:

Thank you for improving our manuscript by your revision. We are greatful to get this positive and encouraging feedback.

Reviewer #2:

The manuscript entitled “3D printing of bioreactors in tissue engineering: A generalised approach” was designed a guidance including test bodies to elucidate the real printing performance for a given printer system. In this manuscript, the authors have done a lot of very meaningful work, performance parameters such as the accuracy and mechanical stability of the test body are analyzed, and post-processing steps such as sterilization or cleaning are also considered.

Response:

Thank you very much for your encouraging feedback and the valuable advices. We appreciate the opportunity to improve our manuscript.

1) What problems should be paid attention to in the design of bioreactor?

Response: 

We fully agree to the reviewers comment. There are many requirements and challenges when designing a bioreactor for tissue engineering applications which are extensively discussed in the literature and cited in the manuscript. We included the following sentence in the introduction (line 68-69):

“Further requirements and general concepts of bioreactor design have been discussed in the literature [23,25].”

Nevertheless, in the introduction (line 59-69) we describe the general requirements for bioreactor systems in tissue engineering. In the results (line 459-469), we point out for what purpose the printed bioreactor was designed and how this is reflected in the concept of the bioreactor. 

To further emphasize a major problem in 3D printing of the bioreactors, we inserted two sentences regarding the discrepancy between the conceptualized and the printed part (606-608):

“A crucial problem is the discrepancy between the conceptualized and the printed parts. According to the gained knowledge about tolerances and limitations of the printer, the designed geometry of the bioreactor was adapted.”

We hope that we could clarify this comment sufficiently.

2) Please explain the difference and connection between bioreactor and bionic.

Response:

Line 78-79: We state, that 3D printing is much better suited to manufacture irregular shapes as they can be found in nature, compared to classic subtracting methods. These shapes offer the option to print more complex bioreactor geometries. We noticed, that the phrase “bionic” could be missleading, so we exchanged the phrase „bionic“ by „organic“ and adapted the sentence accordingly:

„These techniques are well suited for rapid prototyping of complex organic shapes, as they can be found in nature, and hollow geometries.”

3) As we all know, 3D printing still has many problems, including high cost, limited materials, insufficient precision, etc. How to consider and solve these problems in this article?

Response:

We fully agree to your comment. Our publication is intended as a guideline to test printers, technology and materials to simplify the application for non-expert users. However, as every set-up and every printer is different, everyone will have to adjust the process for the own needs and purposes to overcome these limitations (116-130, 546-556).

We can give assistance to choose the best printing method or material (Fig 5), but we can not solve these fundamental problems in this paper.

We hope that we could respond to the comment sufficiently.

4) All figures need to be modified for higher quality.

Response:

Thank you for highlighting this problem for us. We have checked our figures using the Preflight Analysis and Conversion Engine (PACE) und uploaded the respective figure files with higher quality.

5) The quality of English needs improving. The manuscript should be concise and emphasize the most important points of your work.

Response:

Thank you for indicating problems in terms of the language. We had an additional check for the quality of English by a professional English-speaking person, who we also added to the acknowledgements section. All changes made are highlighted in the revised manuscript.

6) Most of the references are relatively old, I recommend the authors to cite new references in recent years.

Response:

Thank you for raising this issue. We carefully revised our literature references and added more relevant and recent ones. 

For most of the older references, we couldn’t think of an adequate replacement. In the introduction, we explain the term Tissue Engineering. As this term was introduced more than 30 years ago, there are many older publications explaining and defining this field. Publications that are more recent do not explain this issue in detail and refer to these older publications. Furthermore, we give examples for tissue engineered substitutes and applications of 3D printing in medicine, which have already been achieved. Therefore, we cite the original publications, which is the common and good practice. 

All in all, more than 50 % of our references were published between 2016 and 2020.

We hope that we were able to address the reviewer’s comment adequately.

We thank the reviewers for the thorough evaluation of the manuscript and the constructive comments.

---

## [Decision Letter · Decision Letter 1]

6 Nov 2020

3D printing of bioreactors in tissue engineering: A generalised approach

PONE-D-20-27307R1

Dear Dr. Gensler,

We’re pleased to inform you that your manuscript has been judged scientifically suitable for publication and will be formally accepted for publication once it meets all outstanding technical requirements.

Kind regards,

Yi Cao

Academic Editor

PLOS ONE

Additional Editor Comments (optional):

Reviewers' comments:

Reviewer's Responses to Questions

**Comments to the Author**

1. If the authors have adequately addressed your comments raised in a previous round of review and you feel that this manuscript is now acceptable for publication, you may indicate that here to bypass the “Comments to the Author” section, enter your conflict of interest statement in the “Confidential to Editor” section, and submit your "Accept" recommendation.

Reviewer #1: All comments have been addressed

Reviewer #2: All comments have been addressed

2. Is the manuscript technically sound, and do the data support the conclusions?

Reviewer #1: Yes

Reviewer #2: Yes

3. Has the statistical analysis been performed appropriately and rigorously? 

Reviewer #1: Yes

Reviewer #2: Yes

4. Have the authors made all data underlying the findings in their manuscript fully available?

Reviewer #1: Yes

Reviewer #2: Yes

5. Is the manuscript presented in an intelligible fashion and written in standard English?

Reviewer #1: Yes

Reviewer #2: Yes

6. Review Comments to the Author

Reviewer #1: The authors have addressed the plethora of comments raised by reviewers in the initial review and the manuscript is acceptable for publication.

Reviewer #2: The authors have improved the manuscript after revision thus I think it can be accepted in this journal.

7. PLOS authors have the option to publish the peer review history of their article (what does this mean?). If published, this will include your full peer review and any attached files.

Reviewer #1: No

Reviewer #2: **Yes: **Zhipeng Gu

---

## [Editor Report · Acceptance letter]

11 Nov 2020

PONE-D-20-27307R1 

3D printing of bioreactors in tissue engineering: A generalised approach 

Dear Dr. Gensler:

I'm pleased to inform you that your manuscript has been deemed suitable for publication in PLOS ONE. Congratulations! Your manuscript is now with our production department. 

Kind regards, 

on behalf of

Dr. Yi Cao 

Academic Editor

PLOS ONE